



# Future heat extremes and impacts in a convection permitting climate ensemble over Germany

Marie Hundhausen[1], Hendrik Feldmann[1], Natalie Laube[1], and Joaquim G. Pinto[1]

[1]Institute of Meteorology and Climate Research, Department Tropospheric Research (IMK-TRO), Karlsruhe Institute of Technology (KIT), Karlsruhe, Germany

**Correspondence:** Marie Hundhausen (marie.hundhausen@kit.edu)

**Abstract.** Heat extremes and associated impacts are considered the most pressing issue for German regional governments with respect to climate adaptation. We explore the potential of an unique high-resolution convection permitting (2.8 km), multi-GCM ensemble with COSMO-CLM regional simulations (1971-2100) over Germany regarding heat extremes and related impacts. We find an improved mean temperature beyond the effect of a better representation of orography on the convection permitting scale, with reduced bias particularly during summer. The projected increase in temperature and its variance favors the development of longer and hotter heat waves, especially in late summer and early autumn. In a 2° (3°) warmer world, a 26 % (100 %) increase in the Heat Wave Magnitude Index is anticipated. Human heat stress (UTCI>32°C) and local-specific parameters tailored to climate adaptation, revealed a dependency on the major landscapes, resulting in significant higher heat exposure in flat regions as the Rhine Valley, accompanied by the strongest absolute increase. A non-linear, exponential increase is anticipated for parameters characterizing strong heat stress (UTCI>32°C, tropical nights, very hot days). Providing local-specific and tailored climate information, we demonstrate the potential of convection permitting simulations to facilitate improved impact studies and narrow the gap between climate modelling and stakeholder requirements for climate adaptation.

## 1 Introduction

The last two decades have been characterised by an increased number of summer heat waves (HWs), some of them of unprecedented magnitude and impact (e.g. Schär and Jendritzky, 2004; García-Herrera et al., 2010; Barriopedro et al., 2011; Russo et al., 2015). HWs are the most visible sign of ongoing global warming in Central Europe (IPCC, 2021), which lead to an increased awareness in our society and stakeholders (Lee et al., 2015; Moser, 2016). As a result, both government agencies and the private sector have not only developed plans for long-term investments towards climate protection, but also for the development of sustainable adaptation strategies, which are are now regularly finding their way into policy agenda (Biesbroek et al., 2010). In Germany, local governments are key actors implementing adaptation strategies (Hackenbruch et al., 2016). Nearly one fourth of the German cities had climate adaptation plans in place by 2018 (Reckien et al., 2018), documenting an increasing interest in the subject. Moreover, the German federal government has launched large research activities like the RegIKlim consortium (regional information for action on climate change) to further strengthen this development.



From the perspective of administrations in municipalities in Southern Germany, the greatest need for action lies indeed in
the adapting to heat extremes (Hackenbruch et al., 2017). HWs – increased temperature over several consecutive days – are a
thread to ecosystems, economy and human health (e.g. Basu and Samet, 2002; Poumadere et al., 2005). HWs are in fact the
weather hazard causing the highest number of deaths in Europe (Zuo et al., 2015). E.g., for the European HW in 2003 alone
up to 80000 additional deaths were recorded accumulated over Europe in the twelve countries concerned by excess mortality
(Robine et al., 2007). However, there is no unified definition of a HW. Different thresholds for e.g. length and temperature
can be found in the literature, and a variety of indices have been developed for classification, e.g. Warm Spell Duration Index
(WSDI) (Alexander et al., 2006), the Heat Wave Magnitude Index (HWMId) (Russo et al., 2014), or excess heat factor (EHF)
(Nairn and Fawcett, 2015). Recent efforts have gone towards quantitative approaches and a higher comparability between
methods (e.g. Perkins and Alexander, 2013; Russo et al., 2014; Becker et al., 2022), leading to a better understanding of the
strengths, weaknesses and range of applicability of the individual indices. Irrespective of index used, there is a clear consensus
in the scientific community (IPCC, 2021) that HWs will become more severe in terms of duration, frequency and magnitude
with increasing global warming, also in Central Europe.

Climate information on the regional to local scale is needed for the development of tailored climate adaption measures. This
can be achieved with regional climate models (RCM), which perform a downscaling of the climate projections from global
climate models (GCMs) to the required spatial and time scales, as it is done in the Coordinated Regional Downscaling Ex-
periment CORDEX (e.g. Jacob et al., 2014). Novel developments include RCM simulations performed with a grid spacing
under 4 km, which resolves convection permitting scales and thus parametrizations of deep convection are not required (con-
vective permitting models, CPM) (Prein et al., 2015). Due to very high resolution on the scale of urban districts, relevant data
fields can be either derived directly or allow a direct coupling with impact models. Several recent studies have documented the
advantages of this convection permitting simulations, both in terms of dominant convective precipitation but also in regions
with strong spatial heterogeneity as present in mountainous or urban areas (Prein et al., 2015). Regarding the representation of
temperature, there is not yet a consensus of added value in convection permitting simulations. Whereas Prein et al. (2013) and
Brisson et al. (2016) attribute improvements of the temperature output the better resolution of orography, Ban et al. (2014) even
found an increasing bias on the convection permitting scale but improvements of the diurnal cycle of temperature in a domain
covering the alpine region. In contrast, an improvement of mean temperature was found in Hohenegger et al. (2008) for most of
her study area and in investigations by Hackenbruch et al. (2016) over Germany. In addition, Tölle et al. (2018) found an added
value for temperature extremes. Mixed results with a regional dependency were found in Soares et al. (2022), concluding a gain
for temperature due to an improved spatial representation of local atmospheric circulations and land-atmosphere interactions.

To quantify the associated uncertainties of the regional climate projections, ensemble simulations are required. As the com-
putational costs in CPM are (very) high, many climate studies are based on single model projections, and only few studies using
CPM ensembles exist (Prein et al., 2015). Very first ensembles of convection permitting climate projections exist e.g. from the
CORDEX Flagship Pilot Study on Convection (FPSConv; Pichelli et al. (2021); Ban et al. (2021)). There, several GCMs from
the Coupled Model Intercomparison Project CMIP5 (Taylor et al., 2012) using the RCP8.5 scenario (Van Vuuren et al., 2011)
were downscaled by multiple RCMs to a common grid with 3 km resolution covering the larger Alpine area (ALP-3). They





used 10-year time-slices for the historical period (1996-2005) and two future periods (2041-2050 and 2090-2099). The current

study applies a different ensemble approach, that is a four-member ensemble of convection permitting climate projections performed by a single RCM, downscaling four GCMs under the scenario RCP8.5. All simulation cover the period from 1971 until 2100 in a quasi-transient manor. To our best knowledge, an ensemble of this temporal extend is currently unique. Such long simulation period allows for a better statistical representation of extremes and the application of approaches used for typical coarser scale transient GCM or RCM ensembles, like e.g. the analysis for different Global Warming Levels (GWL) as it is used

in the IPCC AR6 (Lee et al., 2021) to compare climate change signals for GCMs with different climate sensitivity or between different emission scenarios.

Our focus in this study is heat extremes and related impacts under global warming compared to recent climate conditions. Specifically, we were motivated by three guiding questions:

1. What are the benefits of convection permitting models for temperature extremes in Germany? (Section 3)

2. What can we learn from a convection permitting ensemble about future regional temperature trends and HW characteristics? (Section 4 & 5)

3. What is the impact of these changes on heat stress and other regionally mapped tailored climate parameters? (Section 6)

The paper is structured as follows: Section 2 describes the methodology and the used datasets. Section 3, 4, 5, and 6 focus on the results guided by the three research questions, while a summary and discussion concludes the paper in section 7.

## 2   Data and method

### 2.1   The COSMO-CLM ensemble

The simulations analysed in this study have been generated in the context of the projects KLIWA (Klimaveränderungen und Konsequenzen für die Wasserwirtschaft) and been extended within the project ISAP (Integrative urban-regional adaptation strategies in a polycentric growth region: Model region – Stuttgart Region). The regional climate simulations are conducted

using the RCM COSMO5.0-CLM9 (CCLM, Rockel et al. (2008)). CCLM originates from the German weather service forecast model COSMO (Baldauf et al., 2011), which is a three-dimensional, non-hydrostatic, fully compressible numerical model for the atmosphere including a multi-layer soil-vegetation transfer model TERRA-ML (Schrodin and Heise, 2001). the RCM has been applied for multiple studies over different CORDEX domains (Sørland et al., 2021) and on the kilometre-scale within the CORDEX Flagship Pilot Study on convection (Ban et al., 2021; Pichelli et al., 2021).

Initial and boundary data are provided by four GCMs (cf. Table 2) from the CMIP5 generation under the scenario RCP8.5 (Van Vuuren et al., 2011). The selected GCMs cover a wide range of climate sensitivities (Nijsse et al., 2020). In addition, an evaluation simulation downscaling ERA40 (Uppala et al., 2005) over the period 1971-2000 is included (Hackenbruch et al., 2016), using the same setup as the projections.





**Table 1.** Model setup.

| Nesting level | Grid spacing | Grid dimensions (lat, lon, level) | Remarks |
|---|---|---|---|
| 1st nest | 0.44°, 50 km | $118 \times 110 \times 40$ | Convection parametrized (Tiedtke, 1989) |
| 2nd nest | 0.0625°, 7 km | $160 \times 200 \times 40$ | Convection parametrized (Tiedtke, 1989) |
| 3rd nest | 0.025°, 2.8 km | $322 \times 328 \times 49$ | Only shallow convection parametrized |

The ensemble was generated in a three-step nesting approach (Table 1; Fig. 1a) with a first nest over Europe with 0.44°

grid resolution, an intermediate nest over Central Europa with 7 km resolution an a inner nest that encompasses the area of Central/Southern Germany and the Alpine area with 2.8 km resolution. The convection for the first two nests is parametrized using the Tiedtke-scheme (Tiedtke, 1989). For the inner domain, this parametrization is only used for shallow convection as in Hackenbruch et al. (2016). The first two nesting levels were performed in a transient way. The third nest was originally performed in 30-year time-slices preceded by a three year spin-up (1968-2000; 2018-2050; 2068-2100; Schädler et al. (2018)).

These time-slices were later on extended (2001-2020; 2051-2070) to provide a quasi-transient ensemble for the whole period. The overlapping periods (2018-2020 and 2068-2070) were compared (not shown). No significant climatological differences were found several month after simulation start, in accordance with the findings from Lavin-Gullon et al. (2022).

This continuous time-series data enabled us to apply the concept of Global Warming Levels (GWLs) (Lee et al., 2021) allowing an improved comparability from the downscaling of GCMs with differing climate sensitivities or different emission

scenarios. Therefore, this approach mitigates parts of the GCM and scenario uncertainties and provides more specific information about the effects of climate change given a certain threshold of warming. Specifically, we analyse the +2 K and +3 K GWLs, which was possible for all GCMs due to the use of the high-end scenario RCP8.5. An overview about the simulations is given in Table 2. The period 1971-2000 is used as historical reference period, which is attributed a global warming of 0.46 K. Table 2 lists the 30-year periods for the GCMs, which are centered around the respective year of the threshold exceedance

similar to Teichmann et al. (2018).

As a strong dependency of the temperature output on the major landscape was detected, the area is narrowed down to a geographically more homogeneous area (Fig. 1b) including the Central Uplands, the South German Scarplands and the Alpine Foreland. Therefore, the domain focuses on the hilly parts of Germany, excluding the flat regions – in Northern Germany – and the mountainous regions – the Alps – in the very south. This domain – later referred to evaluation area – is bordered in red in

Fig. 1b and is used in this study when statistics are applied over several grid points. The analysis in the paper is largely focused on HWs and associated impacts in the warm season. Since we observed the largest changes in late summer and early fall, we limit the analysis in this case to the months of May through October. This period is referred to as the summer half-year below.

To evaluate the skill of the convection permitting simulation a comparison of the second, convection parametrizing nest with the third, convection permitting nest is performed, using the gridded observation dataset HYRAS (Rauthe et al., 2013;





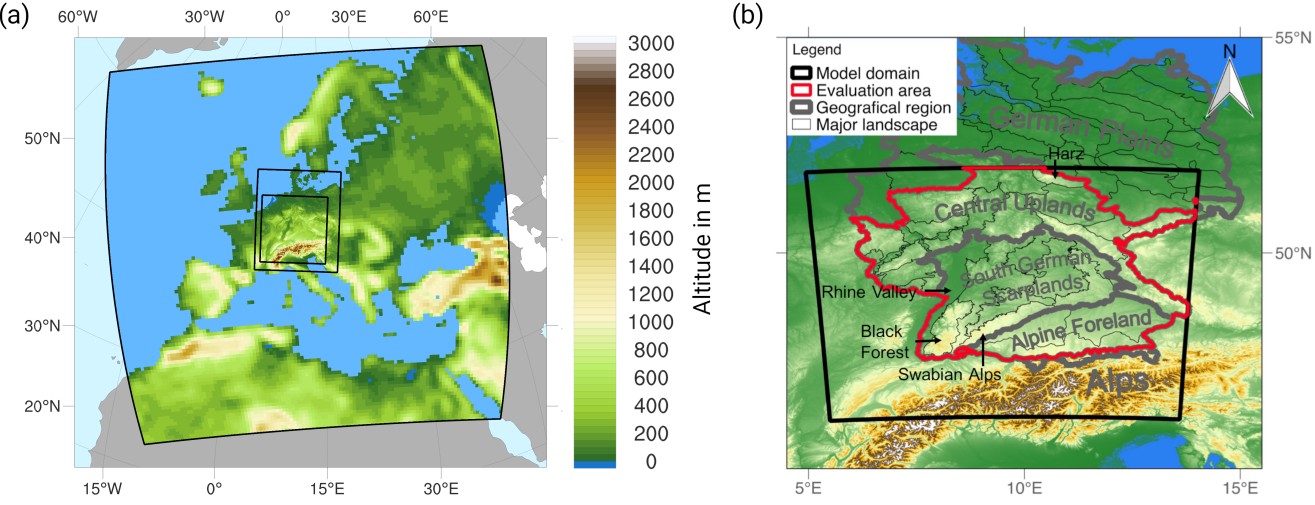

**Figure 1.** Nesting in (a) and model domain, with the sponge area truncated and the used evaluation area red in (b). Important major landscapes for the evaluation are the Rhine Valley, the Black Forest, the Swabian Alps, and the Harz (Shapefiles of the major landscapes: Bundesamt für Naturschutz (BfN), 2015).

**Table 2.** Name, realization, Equilibrium Climate Sensitivity (ECS; cf. Nijsse et al. (2020) supplementary), 30-year periods corresponding to GWL +2 and +3 degrees relative to pre-industrial conditions and main reference for the CMIP5 GCMs downscaled for the ensemble.

| GCM | Realization | ECS in °C | GWL2 | GWL3 | Reference |
|-----|-------------|-----------|------|------|-----------|
| CNRM-CM5 | r1i1p1 | 3.28 | 2029 - 2058 | 2052 - 2081 | Voldoire et al. (2013) |
| MPI-ESM-LR | r1i1p1 | 3.66 | 2029 - 2058 | 2052 - 2081 | Giorgetta et al. (2013) |
| EC-EARTH | r12i1p1 | 4.18 | 2026 - 2055 | 2051 - 2080 | Prodhomme et al. (2016) |
| HadGEM2-ES | r1i1p1 | 4.64 | 2016 - 2045 | 2037 - 2066 | Collins et al. (2011) |

Razafimaharo et al., 2020) as reference. The comparison is conducted for the reference period 1971-2000 for the evaluation run driven by ERA40 as well as for all ensemble members. Simulation data was interpolated on the HYRAS grid with a grid spacing of 5 km. In addition, a height correction of temperature was applied along with the interpolation, assuming a vertical gradient of 0.0065 K/m. The correction compensates the effect of a height dependent temperature that is favored by higher resolution of orography. The evaluation of the model skill was conducted prior to the bias correction.

**2.2 Bias correction**

In order to obtain reliable data and to correct for a systematic error in climate simulations, it is common practice to apply a bias correction (e.g. Gudmundsson et al., 2012). Among the most popular methods is Quantile Mapping (QM). The distribution of





the modeled variable $P_m(x)$ is transformed to a new distribution that equals the distribution $P_o(x)$ of the observed variable in an evaluation period:

$$P_o = h(P_m) \tag{1}$$

The transformation is defined as Eq. (2) using the cumulative density function $F$ of the model output and observation in the evaluation period, indicated with the subscript hist.

$$P_o = F_{o,\mathrm{hist}}^{-1}(F_{m,\mathrm{hist}}(P_m)) \tag{2}$$

We use a parametric-based QM for the correction of temperature as there are indications of a more robust mapping function than in an empirical approach (Lafon et al., 2013). Here $F_o$ and $F_m$ are derived by fitting a distribution to the data, which is the normal distribution for temperature (Berg et al., 2012; Qian and Chang, 2021). In contrast the empirical approach is used for precipitation, as no added value was found with the distribution-based method with e.g. a gamma distribution as in Piani et al. (2010) or Ehmele et al. (2022). In addition, a dry day correction following Ehmele et al. (2022) was applied prior to QM for precipitation.

The bias correction was derived for the parameters daily mean temperature $T_{\mathrm{mean}}$, daily minimum temperature $T_{\mathrm{min}}$, daily maximum temperature $T_{\mathrm{max}}$, and the daily precipitation sum $P_{\mathrm{sum}}$. As reference the observation dataset HYRAS with a resolution of 5 km was used, that was interpolated to the model grid. Along with the interpolation a height correction of $T_{\mathrm{mean}}$, $T_{\mathrm{min}}$ and $T_{\mathrm{max}}$ was applied assuming a vertical gradient of 0.0065 K/m. The available 30 years of the historical time slice from 1971 to 2000 were used as reference period. To account for seasonal dependencies, the bias correction was applied for each month $i$ of the year separately, using a transfer function derived and applied over month $i-1$ to month $i+1$.

As shown in Fig. 2, major improvements can be achieved by the distribution based QM using the example of $T_{\mathrm{mean}}$. In panel (a) the reference data HYRAS is shown averaged over the summer half-year. Comparing this reference data to the simulations driven by ERA40 a root mean square error (RMSE) between 1.95 °C and 2.18 °C (5. to 95. percentile) is visible in the evaluation area (Fig. 2b). The skill of the applied bias correction is expressed over the mean squared error skill score, MSESS, using the mean square error, MSE (Eq. (3)). MSESS is positive all over the domain (Fig. 2c), thus the correction leads to a better alignment of the simulation data with the observation. Stronger improvements coincide with regions of higher deviations of the uncorrected data.

$$\mathrm{MSESS} = 1 - \frac{\mathrm{MSE}_{\mathrm{corr,obs}}}{\mathrm{MSE}_{\mathrm{raw,obs}}} \tag{3}$$

## 2.3 Heat wave and impact indices

Different aspects of heat stress are addressed with this study. We start with the classical approach of describing the meteorological aspects of HWs. Secondly, we will focus on the impact on human health using a thermo-physiological description of heat. Finally, user-oriented parametrizations are tested. All metrics used are presented in the following.





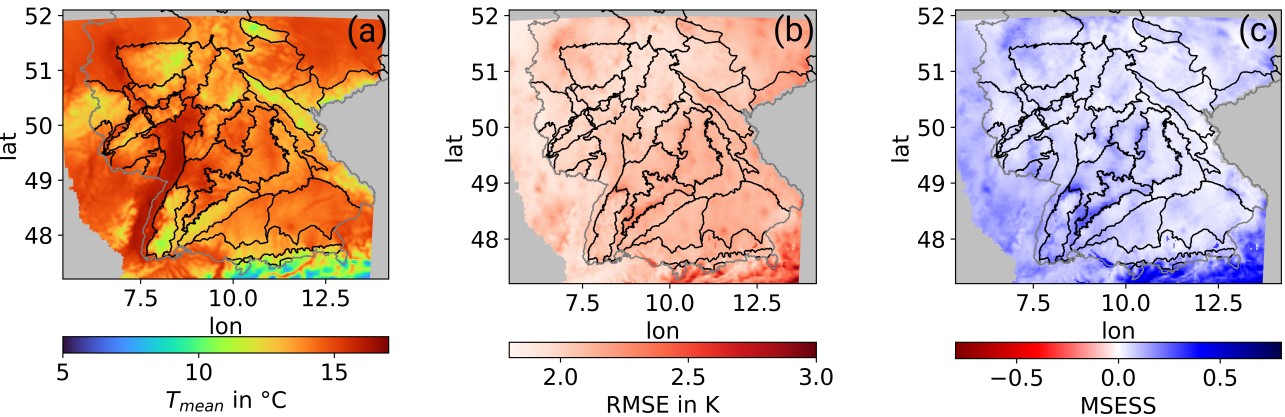

**Figure 2.** Impact of the bias correction of $T_{\mathrm{mean}}$ in the summer half-year (May to October) comparing the ERA40 driven model run with observation. (a) shows the mean summer temperature 1971-2000 in the reference dataset HYRAS, (b) the RMSE of the ERA40 driven model run compared to HYRAS and (c) the MSESS of the bias corrected run compared to the uncorrected.

### 2.3.1 Heat wave indices

A number of consecutive days with elevated temperature is called a HW. However, a universally fitting definition does not exist, but several definitions can be found in literature. We here use the definition by Russo et al. (2014), in which a HW is defined as an uninterrupted series of at least three days where the daily maximum temperature $T_{\mathrm{max}}$ exceeds $T_{\mathrm{max,90\%}}$, the daily 90th percentile of $T_{\mathrm{max}}$ within a 31-day centered window over the reference period. Several metrics describing different aspects of HWs exist. The length of a HW is derived as number of consecutive HW days, its frequency is the average number of HW-days per year. As a measure for the HW temperature we introduce the maximum excess temperature $\Delta T_{\mathrm{max}}$ above the 90th percentile threshold. Russo et al. (2014) proposed a Heat Wave Magnitude Index (HWMId) an index that can be compared across regions and time, taking HW-length as well as temperature into account. The HWMId is calculated over

$$\mathrm{HWMId} = \frac{T_{\mathrm{max}} - T_{\mathrm{max,25\%}}}{T_{\mathrm{max,75\%}} - T_{\mathrm{max,25\%}}} \tag{4}$$

with $T_{\mathrm{max,25\%}}$ and $T_{\mathrm{max,75\%}}$ the daily 25th and 75th percentile of $T_{\mathrm{max}}$ with a 31-day centered window in the reference period. The event sum over the heat event characterizes the magnitude of a HW.

### 2.3.2 Human heat stress

Apart from air temperature, there are additional elements such as clothing, humidity, mean radiant temperature, air movement, and metabolic rate that determine a person's level of thermal comfort (Fanger, 1970). With the requirement to transform this complex system into an application-friendly model, the universal thermal climate index (UTCI) was developed in 2009 from an interdisciplinary collaboration between human thermophysiology, physiological modeling, meteorology and climatology





**Table 3.** Assessment scale of heat stress using the UTCI. Cold stress for UTCI $\leq 9\,^\circ$C is not shown here.

| UTCI in $^\circ$C | Category |
| --- | --- |
| 9 to 26 | no thermal stress |
| 26 to 32 | moderate heat stress |
| 32 to 38 | strong heat stress |
| 38 to 46 | very strong heat stress |
| above 46 | extreme heat stress |

(Jendritzky et al., 2008). The Index is defined as the air temperature of a reference condition causing the same thermal comfort as the actual response. The reference conditions were determined as a wind speed $\mathrm{WS} = 0.5\,\mathrm{ms}^{-1}$ at 10 m height, a mean radiant temperature $T_{\mathrm{mrt}}$ equal to air temperature $T_{\mathrm{air}}$, vapor pressure $p_v$ that represents a relative humidity of $\mathrm{RH} = 50\,\%$. At high air temperatures ($T_{\mathrm{air}} \geq 29\,^\circ$C) the reference humidity is constant at $20\,\mathrm{hPa}$ (Błażejczyk et al., 2013). In Table 3, the defined categories for heat stress are listed.

$$\mathrm{UTCI} = f(T_{\mathrm{air}}; T_{\mathrm{mrt}}; \mathrm{WS}; p_v) = T_{\mathrm{air}} + \mathrm{offset}(T_{\mathrm{air}}; T_{\mathrm{mrt}}; \mathrm{WS}; p_v) \tag{5}$$

$$T_{\mathrm{mrt}} = \left(T_g^4 + \frac{h_{Cg}}{\epsilon d_g^{0.4}}(T_g - T_{\mathrm{air}})\right)^{\frac{1}{4}} \quad , \text{with } h_{Cg} = 1.1 \cdot 10^8 \mathrm{WS}^{0.6} \tag{6}$$

The calculation of the UTCI (Eq. (5)) is based on Fiala's multi-segment model of human physiology and thermal comfort (Fiala et al., 2012), coupled with a clothing model by Havenith et al. (2012). Details can be found in e.g. Jendritzky et al.
(2012); Fiala et al. (2012); Havenith et al. (2012). $T_{\mathrm{mrt}}$ was calculated as in Eq. (6) (Kántor and Unger, 2011), with a wet bulb globe temperature $T_g$ that was approximated according to Liljegren et al. (2008). In Eq. (6) the emissivity is $\epsilon \approx 0.95$ and the diameter of the globe $d = 50\,\mathrm{mm}$. The hourly model results were taken as input for the calculation of UTCI in this study. Due to missing hourly, gridded observations, no bias correction was applied.

### 2.3.3  User-tailored Climate Indices

More and more sophisticated indices were developed focusing on different aspects of heat stress. However, in order to take action in the local governments, the exact information on the change of climatic conditions is not always helpful – on the contrary. The so-called „climate information usability gap" is the barrier about what scientists see as useful and what users consider useful for their decision making. One key aspect to narrow the gap is the customization and tailoring of the data to the user's need to improve the usability of climate information (Lemos et al., 2012), often as a co-design approach.
In the case of climate adaption strategies the measures of interest are according to Hackenbruch et al. (2017) meteorological events leading to an effect on people/health risks (for example, hot days), influence on capital investments or municipal budgets (for example, winter services) or property damage (for example, heavy precipitation events).





**Table 4.** Definition and field of action of the tailored climate parameters related to temperature development based on the KLIMOPASS project (Schipper et al., 2016). $T$ is daily mean, max or min temperature, $T_{\mathrm{JJA}}$ and $P_{\mathrm{JJA}}$ are the mean daily temperature and precipitation sum from June to August. The subscript *clim* refers to the climatological mean, that was calculated over the reference period 1971-2000. The lower limits of $T_{\max}$ for walking weather are: 0 °C for Dez, Jan, Feb, 5 °C for Mar and Nov, 10 °C for Apr, May, Sep,and Oct, and 15 °C for Jun, Jul, and Aug.

| Climate Index | Definition | Field of action |
|---|---|---|
| Very hot days | $T_{\max} > 35\,°\mathrm{C}$ | Road construction: Damage to roads and so-called „blow ups" occur due to strong heating of the road concrete. Health: Decrease in mental and physical performance. |
| Tropical nights | $T_{\min} > 20\,°\mathrm{C}$ | Health: impaired regeneration |
| Growing days | $T_{\mathrm{mean}} > 5\,°\mathrm{C}$ | Conservation: Critical to ecosystem composition and development Forestry: Determines the window of opportunity for forest work Agriculture: Impacts the growing zones for certain crops |
| Dry hot summers and years in between | $T_{\mathrm{JJA}} > T_{\mathrm{JJA,clim}} + 1\,°\mathrm{C}$ & $P_{\mathrm{JJA}} < 0.8 \times P_{\mathrm{JJA,clim}}$ | Agriculture, Forestry: Reduced primary productivity of forest and grassland as well as tree mortality at higher extremes Urban planning: Adaption of tree species and assessment of necessary irrigation. The interval in between hot dry summers is essential for recovery. E.g. 5 years are estimated for tree recovery. |
| Conditions for drosophila suzukii | $T_{\mathrm{mean}} > 10\,°\mathrm{C}$ & $T_{\max} < 30\,°\mathrm{C}$ | Agriculture: Changing climate can influence pests. For each crop and pest, conditions have to be assessed separately. The drosophila suzukii, which is a major pest for fruit production in Central Europe is taken as one exemplary quantity. |
| Walking weather | $T_{\max} < 25\,°\mathrm{C}$ & $T_{\max} >$ variable lower threshold (see table description) | Tourism |

To assess the impact of changing temperature we present several user-tailored climate parameters following Hackenbruch et al. (2017). The selected parameters, their definition, and field of action are summed up in Table 4. All parameters are related to regional temperature changes but cover different fields of action and therefore are of concerns of different stakeholders. The aim of the choice is to show the diversity of the effects of climate change and to present the potential of high resolution climate models for climate adaptation.

## 3 The added value of temperature in a convection permitting ensemble

Evaluating ERA40 driven CCLM simulations compared to the observation, there is a cold bias in the simulations over Germany. Figure 3a shows that in the reanalysis driven simulation the mean monthly temperature over the evaluation domain in the 7 km

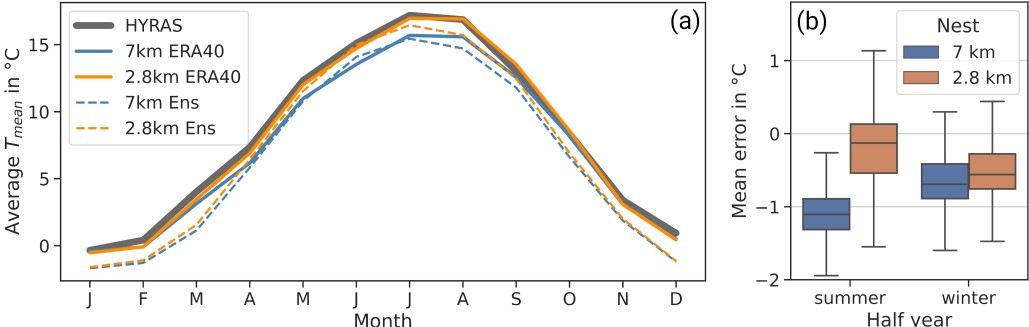

**Figure 3.** Raw output of 2 m temperature in the 2nd (7 km) and 3rd grid (2.8 km) in comparison with the observation dataset HYRAS for the reference period. The analysis was performed on the grid points in the evaluation area. (a) shows the monthly mean temperature in observation, reanalysis and ensemble median. (b) visualizes the mean error of ERA40 time series compared to the observation for summer (May–Oct) and winter half-year (Nov–Apr). The boxplot shows the spread over the grid points.

simulation (blue solid line) is always lower than in the observation (gray solid line). This deviation is larger in the summer months. In the CPM (2.8 km) the monthly temperature is comparably higher in the summer and in autumn even exceeds the observation by 0.5 K. However, there is no strong improvement during the winter months in the mean temperature and the cold bias persists. In the convection permitting ensemble, monthly mean temperature is improved similarly as shown in dashed lines
in Fig. 3a. Again, the largest improvement is in the summer. However, the mean bias in the ensemble median is larger than in the reanalysis, especially in the winter.

Averaged over all grid points, the mean error is reduced from $-1.1\,°C$ to $-0.13\,°C$ in the summer half-year (Fig. 3b). Moreover, the spread is increased. In the winter half-year a smaller, but still significant reduction according to a Wilcoxon signed-rank test with significance level of 0.05 is visible and the median is reduced from $-0.69\,°C$ for 7 km to $-0.56\,°C$ for
2.8 km. Those trends in the temperature output from coarse to high resolution are similar in the ensemble as in the reanalysis driven run. For further information on the performance of the single ensemble members, please refer to the supplementary information (Fig. S1).

To reveal spatial patterns, the mean summer half-year temperature of the 2nd, 7 km nest (Fig. 4a) and the 3rd, 2.8 km nest (Fig. 4b) of the reanalysis driven run are compared to the observation (Fig. 2a). Whereas there is a negative bias at nearly all
grid points for the coarser nest (Fig. 4a), local differences are visible for the convection permitting simulation (Fig. 4b). Here, a negative bias is still present in the north of the domain, especially in the hilly regions. In the south of Germany predominantly positive anomalies are visible. Even though the regions with positive bias are not correlated with altitude, they do not seem to be independent of orography. The largest positive bias is found in the South German Scarplands (lon $\approx 9.0°$, lat $\approx 48.5°$) – located directly between two major mountain ridges, the Black Forest and the Swabian Alps.

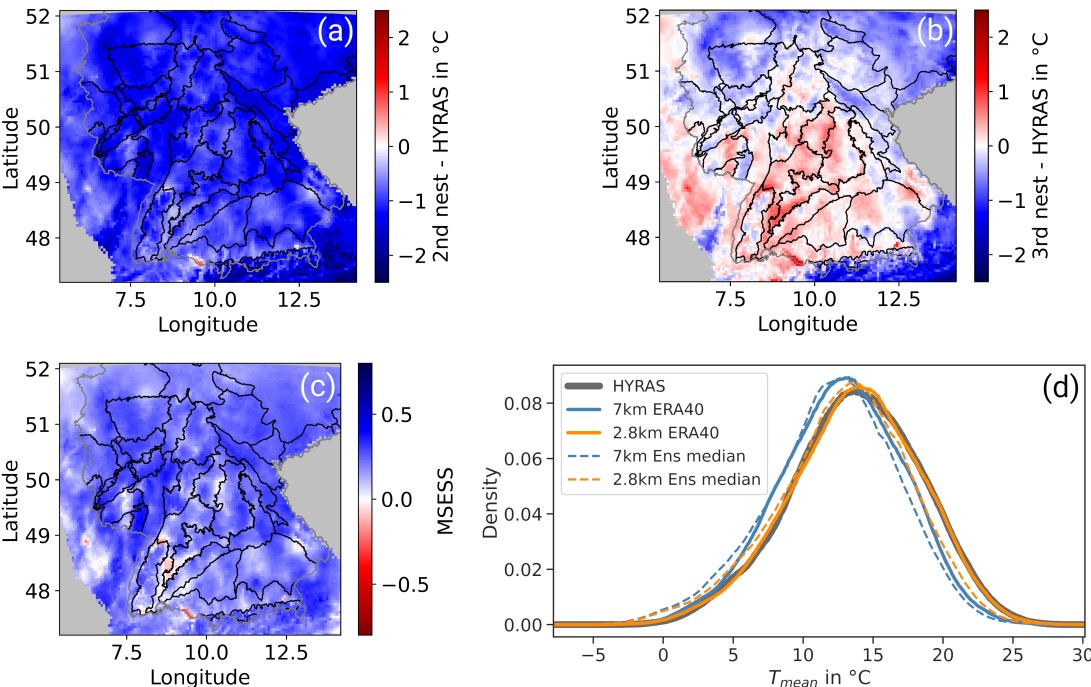

**Figure 4.** Evaluation of ERA40 driven simulation on a convection parametrizing (7 km) and convection permitting (2.8 km) scale for the summer half-year (May–Oct) in the period 1971-2000 compared to the reference observation datas et HYRAS. The difference of the raw output of mean summer temperature is shown (a) for the 7 km simulation and (b) for the 2.8 km simulation. In (c) the MSESS of 2.8 km compared to 7 km is mapped and (c) displays the density distribution of 2.8 km and 7 km in the evaluation area for the reanalysis driven run (solid lines) and the median of the ensemble in the reference period (dashed lines).

For nearly all grid points there is an improvement with the convection permitting simulation which is indicated by a positive MSESS in Fig. 4c comparing the second and third nest with respect to the reference dataset HYRAS. There are few grid points with negative MSESS. Those are associated with a positive bias an an overshoot of the convection permitting simulation.

The density distribution of daily summer temperature shows nearly perfect agreement of observation and the convection permitting reanalysis run (Fig. 4d). In comparison, the distribution for the reanalysis driven 7 km simulations is shifted towards
colder temperatures and has a lower spread. Especially the highest summer temperatures are better resolved by convection permitting simulations. An improvement is also visible for the 2.8 km median of the ensemble simulations compared to the 7 km output. However, especially the high summer temperatures are still underestimated by the CPM. Low temperatures, from approximately $-3\,°C$ to $10\,°C$, are overestimated.

Overall, we identify a significant reduction of the mean bias for the convection permitting resolution, which is especially
pronounced during summer. Over Germany, the convection permitting simulation reproduces a realistic frequency distribution





of daily 2 m temperature. The remaining mean errors shows a trend from negative bias in the north towards positive bias in the south. Other local patterns are partly associated with the predominant landscape regions.

## 4   Regional temperature trends

**Annual cycle** Future temperature is not expected to develop evenly over the year. In the study area the smallest increase is

observed in spring, largest in the late summer and during the winter (Fig. 5a). The behavior is similar for GWL2 and GWL3. The stronger late summer increase leads to a shift of the summer peak of maximum temperature by 12 days in GWL3 compared to 1971-2000.

A closer view in the ensemble spread shows, that throughout the year there seems to be good agreement within the three simulations driven by EC-EARTH, MPI-ESM-LR and CNRM-CM5. There is an average variance of $0.6\,\mathrm{K}^2$ for the mean

temperature over the study area in GWL3. In contrast, warming – especially in the winter and autumn – is significantly more pronounced in the simulation driven with HadGEM2-ES (Fig. 5a, dotted line). Averaged over the year, the temperature increase is 1.5 K higher than for the other simulations by GWL3. HadGEM2-ES is the member with the highest climate sensitivity of the driving GCM within this ensemble (cf. Nijsse et al. (2020); Table 2). In the following, the presented results of HadGEM2-ES will stand out repeatedly as it appears that the nature of its projected climate change signal differs from that in the other three

ensemble members EC-EARTH, MPI-ESM-LR, and CNRM-CM5 with lower climate sensitivity.

**Temperature distribution** Figure 5b shows the density of the daily mean summer temperatures over the evaluation area. In the evaluation period the distribution is skewed left and the peak is at 14.2 °C. The shape of the distribution is reproduced well compared to the observation, however, the ensemble overestimates the probability at the peak of the distribution. In a warmer world, the mode shifts to higher temperatures that are 15.4 °C in GWL2 and 16.6 °C in GWL3. Moreover higher

maximum temperatures up to 27.4 °C (99th percentile) in GWL3 are reached. There is a decline in temperatures left of the peak, respectively. However, especially for low temperatures the magnitude of decrease is relatively small, leading to an increased width of the distribution. As shown in Fig. 5c the full width of half maximum (FWHM) in the ensemble average increases from 10.4 to 12.1 °C. Three out of four ensemble members agree on a steady increase of the width. Only the simulation run by HadGEM2-ES does not confirm an increase of FWHM in the period from 1971-2000 to GWL2.

**Spatial patterns** The average summer temperature (May–Oct) varies significantly over the evaluation area as already shown for the observational data in Fig. 2a and provided in the supplementary information (Fig. S2) for the ensemble mean. It ranges from 12.3 to 15.5 °C (5. and 95. percentile). As expected, highest temperatures are found at low altitudes. The Rhine Valley stands out with highest average temperatures up to 16.6 °C. Lowest average temperatures are accordingly observed in complex regions with pronounced orography: Examples are the Harz (average 12.5 °C) in the Central Uplands or the Black Forest

(average 13.0 °C) in the south. Moreover, spatial heterogeneity is increased in those complex regions.

The summer temperature increases over the whole evaluation area with global warming. From the reference period (Global warming at 0.46 °C) to GWL2, the increase is in average 1.55 °C (Fig. 6a). All in all, warming is rather uniform with a range from 1.45 to 1.64 °C (5. and 95. percentile). The strongest increase is observed in the north of the study domain, in the uplands

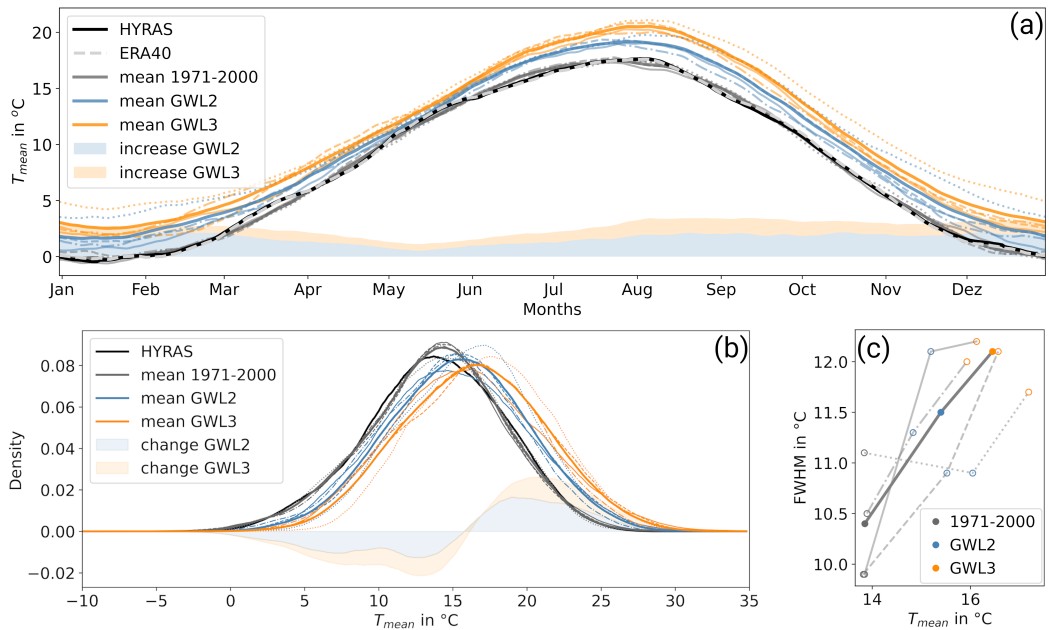

**Figure 5.** The different aspects of the evolution of daily mean temperature $T_{\mathrm{mean}}$ from reference period (gray) over GWL2 (blue) to GWL3 (orange) are shown. (a) displays its annual cycle averaged over the study area and over a 31-day running window. (b) shows the density distribution of daily mean temperature in the summer half-year (May–Oct) and (c) its full width at half maximum (FWHM). Different line styles correspond to different driving GCMs – solid: MPI-ESM-LR, dashed: EC-EARTH, dash-dotted: CNRM-CM5, dotted: HadGEM2-ES, the thick lines correspond to the ensemble mean.

and farther in the south in the Black Forest and Swabian Alps. Less warming is expected in the Alpine Foreland and in the

flatter regions of the South German Scarplands. The ensemble spread seams only partially dependent on the orography and landscape. Moreover, data show a trend superimposed from northwest to southeast with decreasing spread (Fig. 6b).

In GWL3, summer temperature increases further by 2.44 to 2.76 °C (5. and 95. percentile) compared to the evaluation period (Fig. 6c). The spatial patterns of summer temperature increase remain similar with slightly higher increase in hilly regions. Especially in the south, the uplands (Black Forest and Swabian Albs) show an increased warming compared to the

surrounding flatter regions. The ensemble spread in the projection of temperature in the distant future has slightly widened (Fig. 6c). It ranges from 1.12 to 1.48 °C (5. and 95. percentile). It is especially high in the north west of the domain and in areas with higher elevation. Lowest spread is visible in the flat Rhine Valley. The higher deviations at the locations of the large lakes in Southern Germany (Lake Constance (lon 9.4; lat 47.6), Lake Ammersee (lon 11.1; lat 48.0), Lake Starnberg (lon 11.3; lat 47.9), Lake Chiemsee (lon 12.5; lat 47.9)) are caused by interpolation of the water surface temperatures from the coarse grid,

since no lake module was applied.


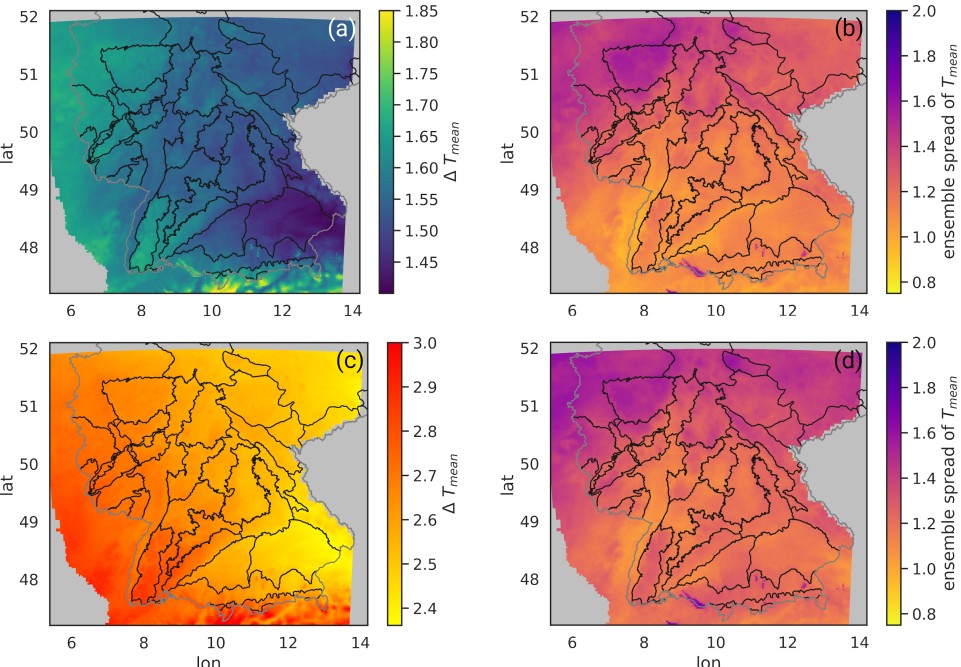

**Figure 6.** Mean development of $T_{\mathrm{mean}}$ in the summer half-year (May–Oct) as ensemble mean compared to the reference period for GWL2 (a) and GWL3 (c) and the according ensemble spread calculated as range between minimum and maximum prediction for each grid point in GWL2 (b) and GWL3 (d).

Overall, the mean temperature over Germany rises in a warmer climate predominantly in late summer as well as in the winter half-year, with the smallest increase in spring. This leads to a general shift of the summer maximum temperatures to later summer. The increase is spatially largely homogeneous, with slightly stronger warming expected in mountainous regions. Moreover, the temperature distribution in a warmer climate is expected to be wider (larger variability), indicating that extreme

temperatures will experience a greater change compared to the average warming.

## 5    Heat wave characterization

This section characterizes HWs in the future based on their different features – frequency, length, temperature, and magnitude. Be aware that throughout this section we are focusing on a relative definition of these events – an anomaly versus the 90th percentile from the reference period (Sec. 2.3.1). The relationship between HW magnitude, duration, and excess temperature is

examined for the most severe HWs in each year in terms of the cumulative HW magnitude in the evaluation area (Fig. 7). The according figure providing absolute HW temperatures is provided in the supplementary (Fig. S3). In Fig. 7a to c, the variance



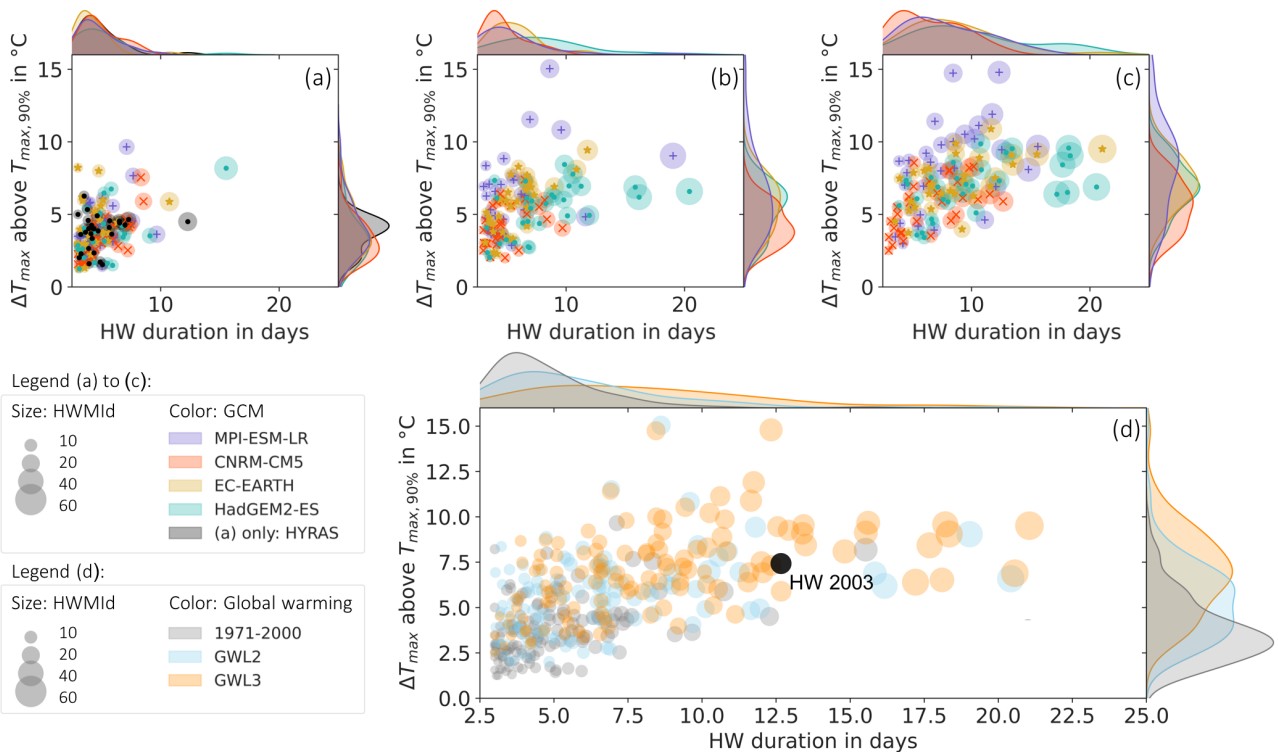

**Figure 7.** Bubble plot of the strongest HW in each summer half-year (May–Oct) in every projection run. The considered events are identified over the maximum of the HWMId integrated over the area. The HW-features of all involved grid points are averaged. Panel a to c show the single ensemble members and comparison with the observation: a for 1971-2000, b for GWL2, and c for GWL3. In Panel d the ensemble members of one GWL are merged to display the evolution over time. The black data point corresponds to the HW in 2003 derived from HYRAS data.

between the ensemble members is studied. Observed average HW duration of the strongest HWs a year ranges from 3 to 12 days. The temperature excess $\Delta T_{\max}$ above the 90th percentile ranges from 1.5 to 6.3 °C. The excess temperature does not correlate strongly with the duration ($r = 0.22$) but the longest HWs are usually associated with high temperatures. The observed HWMId has a range of 5 to 22 with an average of 8.8. HWMId increases with average HW length ($r = 0.99$). The ensemble of climate simulations shows no significant deviation from the observed distributions in the reference period for the characteristics duration and HWMId, confirmed by a two-sample Kolmogorov-Smirnov test on the α level of significance of 0.05. For the excess temperature the test results support no significant deviation from the observed distribution for three out of four ensemble members but significant differences for the results of the CNRM-CM5 driven simulation. The underestimation of the modelled excess temperature is shown in Fig.7a. Moreover a peak around $\Delta T_{\max} = 4$ °C is visible in the observation that is not reproduced by any ensemble member.





The spread between the ensemble members widens over GWL2 (Fig. 7b) to GWL3 (Fig. 7c). CNRM-CM5 develops lowest excess temperature, shortest HWs in the ensemble, and therefore lowest HWMId. The pattern agrees with the underestimation of temperature in the reference period. Hence, the trend seems to persist in the future. For MPI-ESM-LR, the projection leads

predominantly towards higher HW temperatures with a maximum excess temperature of 15 °C in GWL2 and GWL3 but HW duration is about average. EC-EARTH shows an average distribution of length and temperature compared to the other members of the ensemble. In contrast, HadGEM2-ES shows long HW duration with average excess temperature, leading to highest HWMId in average. Discrepancies in HW duration indicate different dynamics in the driving models. In fact, HadGEM2-ES is described as one of the best performing CMIP5 GCMs in past climate for weather types (Perez et al., 2014) and blocking,

which is underestimated in CMIP5 models in general (Brands, 2022). Presented extremely long HWs should therefore not be discounted as an outlier. All ensemble members agree on an increased width of the temperature and duration distribution compared to the reference period.

Superimposing the three GWLs, a clear picture of HW intensification emerges (Fig. 7d). Following trends for duration, temperature and HWMId are derived. Significance of all trends is confirmed by a two-sample Kolmogorov-Smirnov test on the

a level of significance of 0.05.

– There is a shift towards higher **HW excess temperatures** up to 5.3 (GWL2) and 6.9 °C (GWL3) as median. With a HW temperature in the reference period that ranges from 2.6 °C to 4.5 °C (25 and 75 % confidence interval) hardly any HW are occurring today, that will be a common scenario in the future.

– For the **HW duration** there is a shift towards longer HWs. The average increases from 4.3 (reference) over 5.1 (GWL2)

to 7.5 days (GWL3). Moreover, the spread increases drastically which leads to maximum duration up to 21 days.

– **HWMId**, which is mainly correlated with HW duration, shows an increasing spread. A 26 % (100 %) increase in the median of HWMId is expected from reference to GWL2 (GWL3) (reference: 8.2, GWL2: 10.3, GWL3: 16.5).

In order to put the results into perspective, they are compared with an actual reference event for Germany – the HW in 2003. A HW with a strong economic, environmental impact and cause of thousand deaths, referred to as a record HW (e.g., De Bono

et al., 2004). Performing an analog analysis on 2003 HYRAS data, this HW had an average length of 12.7 days, maximum excess temperature of 7.4 °C and HWMId of 26.7. It is visualized in black in Fig. 7d. As expected the event is extremely unlikely in the reference period. Only one simulated summer in the reference period by HadGEM2-ES exceeds the measured event in 2003. In a warmer world, events with such a strength occur with higher probability. In GWL3 such an event is in the 25 % confidence interval of 5.3 °C to 8.4 °C . For duration, the HW 2003 exceeds the 25 % confidence interval of 5.1 to 10.4

days in GWL3 and its duration is ranked 16th in the projections of $4 \times 30$ years, corresponding to a 8-year return period in GWL3. For HWMId its rank of 21 in GWL3 leads to a 6-year period. It should be noted that in this case no distinction is made between ensemble members. The variations between ensemble members discussed earlier indicate the range of uncertainty in this projection. Moreover the analysis considers only the local observations of 2003 limited to the simulation area. Summing up, an event like HW 2003 will become more likely, but is projected to stay an extreme event with a return period of 5 to 10

years.



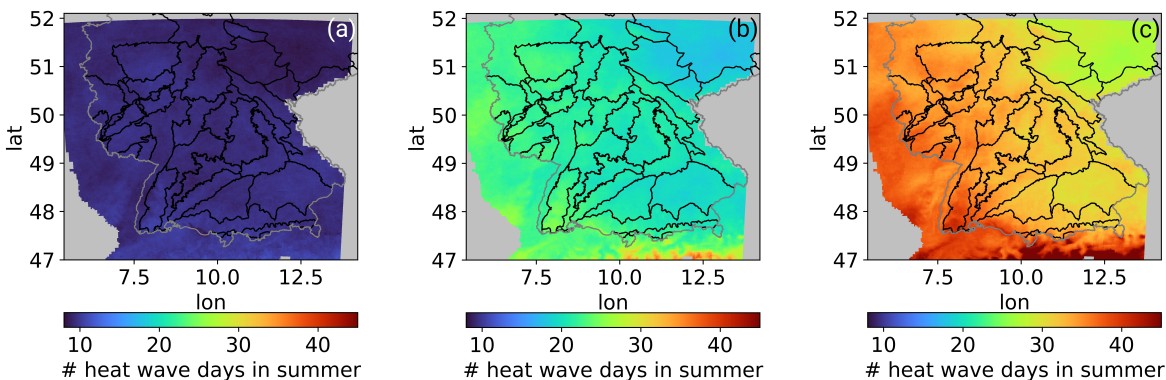

**Figure 8.** The ensemble mean of average number of HW days per summer half-year (May–Oct) in 1971-2000 (a), GWL2 (b), and GWL3 (c).

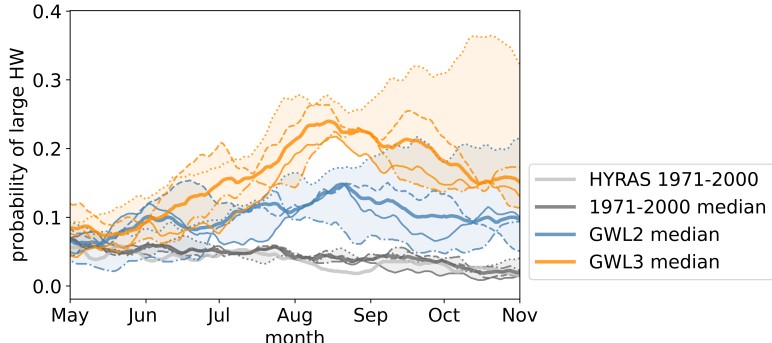

**Figure 9.** Probability of large HW (coverage ≥ 50 %) over the summer (May–Oct) calculated over a 31-day running window. Different line styles correspond to different driving GCMs – solid: MPI-ESM-LR, dashed: EC-EARTH, dash-dotted: CNRM-CM5, dotted: HadGEM2-ES.

To assess regional patterns, the cumulative number of HW days as measure of frequency is analyzed (Fig. 8). Again the summer half-year is considered only. In the reference period, averaged over 30 years, few HW days are observed per summer half-year. In the evaluation area, the number of HW days ranges from 8.7 to 9.9 (5th to 95th percentile) and is distributed relatively uniform across space. An overall increase to 18.8 to 23.0 (5th to 95th percentile) HW days is simulated in GWL2. With even more warming in GWL3, spatial features become visible. The increase affects predominantly the southwest. Moreover slightly enhanced HW occurrence is projected in regions with higher elevation as the Black Forest. Over the domain 28.7 to 36.9 (5th to 95th percentile) HW days are expected in GWL3.

The analysis of the seasonal changes reveals that HW severity is distributed inhomogeneously over the summer (Fig. 9). In the reference period the occurrence of large HWs, defined as HW with a coverage of at least 50 % of the study area, is relatively


flat distributed. There is a declining trend of the probability throughout the summer. From GWL2 to GWL3, it is apparent that there is an increased HW occurrence in late summer, around August and September. All members of the ensemble agree on this trend. However, the magnitude and timing of the adjustment varies. The highest HW probabilities are projected by EC-EARTH and HadGEM2-ES. Some ensemble members even depict a decrease in the occurrence of large HW in early summer in GWL2 (May to June). An intensified HW season especially in late summer, is consistent with the annual cycle of temperature increase

discussed above.

In summary, future HWs are characterised by significantly higher temperatures and longer HW duration. Thus, the magnitude of HWs increases dramatically in a warmer future, namely by 26 % (100 %) in GWL2 (GWL3). Furthermore, enhanced variability is projected for the HW characteristics. While the increase of HW days is spatially largely homogeneous, there is clear seasonality, with a strong increase in HW occurrence in late summer.

## 6   Impacts of temperature and heat increase

The meteorological perspective leaves open the question of the impacts of heat extremes, which will be addressed in the following section. The focus is first on human heat stress, then the analysis is extended to further heat-related climate parameters.

**Human heat stress** The number of days with UTCI $> 32\,°C$ are defined as days with strong human heat stress. As outlined in methods, UTCI is derived from hourly data. Due to missing gridded hourly observations, it was not subjected to bias

correction. As a consequence, the ensemble spread in UTCI is large. Whereas three ensemble members agree on similar range, the simulation of HadGEM2-ES simulates significantly higher numbers of HW days correlated with higher uncorrected summer temperature and seems to be an outlier. Therefore the median of the ensemble for every grid point is displayed in Fig. 10. In the reference period hardly any days per year with strong heat stress are found. The range over the evaluation period is 0.0 to 0.6 days per year (5 to 95 % confidence interval). A maximum number of up to 2.0 days per year averaged over the

reference period in flat regions is visible in the ensemble.

The average number of days with strong heat stress rises in the future GWL2 all over the domain – in average by 0.6 days per year – but with notable spatial differences. Again highest numbers of heat stress days are in the flat Rhine Valley with up to 5.1 days per year (Fig. 11a). Moreover, this region shows the strongest increase from reference to GWL2 which is in average 1.8 days per year. For GWL3, this pattern intensifies with an non-linear, rather exponential increase with global warming (Fig. 11a).

Up to 10.7 days per year with strong heat stress are projected in the hottest region. Also in regions with higher elevation there is a significant increase of future heat exposure, in average 2.3 days are for example expected in the Black Forest by GWL3. For comparison, this exceeds the heat stress that prevailed in the mild, flat Rhine Valley during the reference period.

**User-tailored climate parameters** The analysis of the six tailored climate parameters shows how changing temperature affects further fields of action (Fig. 11b-g). To visualize regional effects, results of two German landscape regions are added

to the graph in addition to the entire the evaluation area: the flat and warmest region of the model domain, the Rhine Valley



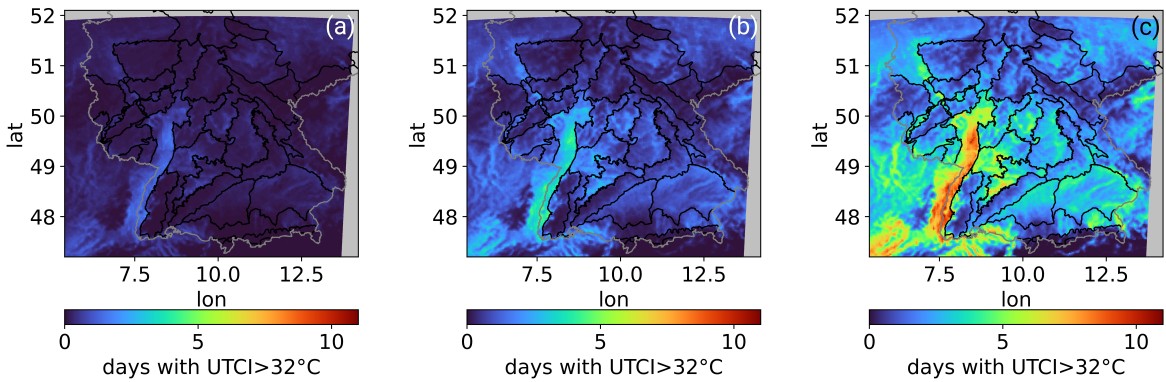

**Figure 10.** Ensemble median of number of days per year with strong human heat stress defined by UTCI > 32 °C for 1971-2000 (a), GWL2 (b), and GWL3 (c).

(dotted boxes) and as counterpart, striped boxes show the Black Forest region, which is geographically directly adjacent to the Rhine Valley and, as a low mountain range, has a high altitude and complex orography (cf. Fig. 1).

Most drastic changes of the mean values are projected for very hot days, tropical nights and dry hot summers (Fig. 11b, c, d). In addition, for very hot days and tropical nights a non-linear, rather exponential increase with global warming is projected. This coincides with a significant increase of variance. The behavior of very hot days and tropical nights is comparable to UTCI (Fig. 11a), implying that this amplified, non-linear increase might be preferentially associated with strong heat stress. The pattern is observed for all shown landscape regions. Differences appear in the absolute values: Heat stress is especially pronounced in the Rhine Valley at low altitude where it exceeds the values in the adjacent Black Forest by a factor of 3.7 (very hot days) or 2.8 (tropical nights) in GWL3.

An about mean linear increase with global warming is visible for dry hot summers and growing days (Fig. 11d and e). Growing days are expected to increase in average by 39 (evaluation area), 40 (Black Forest), 37 (Rhine Valley) days from reference to GWL3, indicating that dependency of the change signal on the region is negligible. Existing regional patterns and variability within a landscape region appear to be preserved in a warmer climate and mean values are subjected to a shift only.

The probability of dry hot summers increases approximately linear as well, accompanied by increasing spatial variance. Here, largest increase is observed in the Black Forest, the smallest in the Rhine Valley. Overall, the probability of a dry hot summer increases drastically: From the reference to GWL3 projected mean increase corresponds to a factor of 4 (Evaluation area and Black Forest) or 3.2 (Rhine Valley).

The two remaining parameters are examples designed for specialized applications in individual, often region-specific challenges – here walking weather for tourism strategy (Fig. 11f) or pests e.g. drosophila suzukii for agricultural planning (Fig. 11g). Using days with walking weather as an example, their number increases in the Black Forest and the variability decreases. The trend in the Rhine Valley is opposite, a decreasing number of days with walking weather with increasing variance. Hence in GWL3 relatively similar numbers of days are to be expected in the two contrasting regions. Also for days with conditions for



**Figure 11.** Ensemble median of UTCI and climate parameters over global warming. The global warming level of the reference period is assumed to be 0.46 K based on Teichmann et al. (2018). The empty box plot visualizes the distribution over the evaluation area, striped boxes represent results in the Black Forest, and dotted in the Rhine valley.

drosophila suzukii, no common trend can be identified and the examples show a climate change signal that depends crucially on local conditions. Such behaviour is mainly attributed to the more complex definition of the parameters with an upper and a lower limit. The evaluations indicate that the more complex the parameter – or the underlying challenge in climate adaptation – the more important the regional consideration becomes.

We conclude that the changes regarding UTCI and user-tailored climate parameters do not necessarily scale linearly with global warming. An over-proportional increase of the climate parameter with global warming is preferably the case for parameters that describe strong heat stress. Moreover, the change signal of climate parameters depends crucially on the landscape





region. In particular for parameters describing strong heat stress, the absolute change signal is highest in flat regions that are already exposed to the greatest heat today. For specialized applications, parametrized over more complex climate parameters, region-specific trends are expected.

## 7   Discussion and conclusion

In the presented analysis of heat extremes and related impacts in a convection permitting climate ensemble for Germany we

could draw three main conclusions:

1. We found an added value for simulated temperature in the convection permitting ensemble especially for hot temperatures, that goes beyond better representation of the topography only. The improvement is particularly prominent in the summer half-year.

2. Mean temperature in the warm season in Germany increases largely homogeneous in space. An increase in temperature

410       variability is found in future projections, which favors the development of longer and hotter HWs especially in late summer. Heat wave magnitude is expected to increase by 26 % (100 %) in GWL2 (GWL3).

3. The changes in human heat stress (UTCI) and tailored climate parameters show a clear dependence on the major landscapes. Heat stress is particularly prominent for low land areas like the Rhine Valley. An over-proportional increase of parameters associated with strong heat stress is found. For the change signal of more complex tailored climate parameters

415       linear behaviour and/or strong dependency on the landscape can be identified.

Our results show an improved representation of 2 m-temperature for CPM due to a reduced cold bias in CCLM. The improved results cannot be attributed solely to the temperature's altitude dependence, which is better represented by higher resolution of orography. This confirms the findings by Hackenbruch et al. (2016), Hohenegger et al. (2008), and Laube (2019). Moreover, the results show that the improvement is largest in the summer (smaller cold bias). However, recent studies have shown that this

temperature bias, especially in daily minimum and maximum, can still be addressed in CCLM with an improved formulation of the 2 m temperature in the land surface scheme (Schulz and Vogel, 2020). Moreover, it needs to be clarified whether the improved temperature output in convection permitting simulation justifies the higher computational cost for high-resolution simulations. While systematic biases between model temperature output and observations remain in our CPM ensemble, we show a clear benefit from a relatively large simulation area across different landscapes. We find a dependency of the remaining

error on the landscape type and an association with orography – especially in transition areas between different major landscape types. Therefore we support a region-specific magnitude of the added value as in Soares et al. (2022). In general, climate change studies focusing on high temperatures and the effects of increasing heat stress are expected to benefit from the better representation of high temperatures and the associated lower impact of bias correction.

The analysis allows for the first time a very high resolution projection of temperature and temperature extremes over Ger-

many in a 2 and 3 degree warmer world. The regional, high resolution analysis confirms general warming over the whole region and a slightly higher change signal in mountainous regions. As in Vautard et al. (2014), the smallest temperature increase was





found in spring. Indeed, the peak of summer temperatures in a warmer climate shifts to later in the summer. Moreover, the analysis confirms a wider distribution of temperature with global warming, implying a greater change of extreme temperature compared to the average warming in the future (e.g. Mearns et al. (1984); Schär et al. (2004); Giorgi et al. (2004); Kjellström
et al. (2007); Vidale et al. (2007); IPCC (2021)). Our study shows, that HW probability is expected to increase significantly over Germany and especially in late summer large HWs are anticipated. HW severity is projected to rise dramatically, indicated by a 26 % (100 %) percent increase from 1971-2000 to a 2° (3°) warmer world. Increasing variability in HW characteristics is projected for the future. This is consistent with past trends of HW temperature and duration derived from observational data (Della-Marta et al., 2007). Our study thus suggests that the trend is likely continue in the future.

Apart from meteorological insights, a closer look at human heat stress and other tailored climate parameters shows the potential of using convection permitting simulations in different fields of application and highlights the importance of individual consideration. Strong human heat stress – parameterized via UTCI>32° as well as associated with very hot days or tropical nights – is prevalent in the flat regions such as the Rhine Valley. Moreover, the largest absolute increase is expected for these regions, comparable to Brecht et al. (2020). The change signal of tailored climate parameters does not always scale linearly
with global warming – as is the case for the relative quantity dry hot summers or growing days, a quantity that targets for moderate conditions. Especially for extreme heat stress (UTCI>32°, very hot days, or tropical nights) we see a non-linear but rather exponential increase with global warming. In particular for specialized applications – expressed e.g. over more complex climate parameters – behaviour depends crucially on the prevailing landscape and might even lead to opposing trends. Therefore, the analysis supports previous results of spatial patterns (Schipper et al., 2019; Brecht et al., 2020) and shows the
benefit of CPM, which allows the representation of distinct characteristics in clearly defined areas.

    Limitations of the study are that the assessment of uncertainty is restricted with four GCMs and only one RCM. However, the magnitude of the uncertainty associated with the RCM choice is typically smaller than from the large scale GCM forcing (Kjellström et al., 2011). In the future, larger ensembles on the convection permitting scales are expected to be available, enabling assessment of GCM- and RCM-uncertainty. Currently, ongoing downscaling of the CMIP6 GCMs are a promising
source of future driving data for high resolution climate simulations. In particular, the improved representation of northern hemisphere blocking in the new generation of climate models (Schiemann et al., 2020) will necessitate additional analysis of HWs and is anticipated to provide complementary insights to the results shown. Moreover, long convection permitting projections would profit from the implementation of variable land surface characteristics over time, as e.g. recently provided by FPS-LUCAS (Hoffmann et al., 2021). Moving from constant to variable input fields could yield valuable information for
heat stress in impact studies. Especially for climate adaptation studies development is still anticipated for urban areas and the evaluation of according urban parametrization schemes. Since no parametrization is used in this study, further improvements for urban areas are to be expected (e.g. Trusilova et al., 2016; Daniel et al., 2019).

    Heat extremes and related impacts derived from a convection permitting ensemble document that the climate change signal depends on major landscape regions. Therefore, such convection permitting projections have the potential to facilitate tailored
impact studies and can help to narrow down the gap between climate research and the requirements of stakeholders e.g. for sustainable risk management and climate adaptation. This presented finding stresses the need of climate adaptation strategies





on a local level and supports the regional approach in climate adaptation research e.g. in the BMBF RegIKlim project: basic research is done in a pilot region, concentrating on region-specific key issues to develop, evaluate, communicate, and test the implementation of adaptation strategies with the aim of an up-scaling in the concerning region in the future.

*Data availability.*   The ensemble simulation data can be requested from the authors. It is planned to provide parts via the German Climate Computing Center (DKRZ). The observation datset HYRAS is a product of the Deutscher Wetterdienst DWD. It can be requested at DWD for research purposes

*Author contributions.*   The concept of the paper was developed by MH, HF and JGP. NL performed preliminary analysis. MH is responsible for data analysis and figures. The initial draft was written by MH, supported by HF and JGP. All authors contributed to discussions, comments,
and revisions.

*Competing interests.*   The coauthor JGP is a member of the editorial board of Natural Hazards and Earth System Sciences.

*Acknowledgements.*   The work was conducted within the funding measure „Regional information for action on climate change" (RegIKlim) of the German Federal Ministry of Education and Research (BMBF) in the ISAP project (01LR2007B) and NUKLEUS project (01LR2002B). The study was carried out in cooperation with the project climXtreme – also supported by the BMBF – at KIT (01LP1901A). We thank Hans-
Jürgen Panitz and Regina Kohlhepp from KIT/IMK for their help in the RCM simulations. The simulations were performed on the national supercomputers Cray XC40 Hazel Hen and HPE Apollo Hawk at the High Performance Computing Center Stuttgart (HLRS) under the grant number HRCM (ID 12801). Parts of the processing and analysis were performed at DKRZ. We thank the Deutscher Wetterdienst (DWD) for providing observational data. Finally, we thank the open-access publishing fund of KIT.



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
