# Peer review of "Future heat extremes and impacts in a convection permitting climate ensemble over Germany"

_Natural Hazards and Earth System Sciences, 2022_

## Author Comment (AC1)

**Anonymos Referee 1:**

The present manuscript analyses temperature extremes and associated impacts through a high-resolution convection-permitting (2.8 km), multi-GCM ensemble with COSMO-CLM regional simulations from 1971 to 2100 over Germany. The study points out a projected increase in temperature and its variance combined with hotter and more lasting heat waves. The analysis also considers a comprehensive set of heat stress and user-tailored climate indices.

The research is surely relevant in terms of temperature extremes analysis and also for considering for the first time a multi-GCM ensemble of convection-permitting (CP) climate simulations over a multi-decadal time period.

The topic is surely fitting with the scope of the journal but different relevant aspects mainly pertaining to the methodological choices and technical aspects related to the simulations performed and analyzed in the study deserve clarification before being reconsidered for publication in NHESS journal.

A: First, we would like to thank the reviewer for her/his insightful comments, which have greatly contributed to improving the text. In making corrections, we have tried to follow the suggestions as closely as possible.

**General Comments**

**General Comment 1:**
My first concern regards the very basis of the numerical simulation strategy adopted. I refer to the three-step nesting dynamical downscaling. It is well-known (e.g., Rummukainen 2010), the importance of a buffer or sponge zone of several grid nodes width between two nesting boundaries. This relaxation zone has the fundamental role of bringing the model solution towards the lateral boundary condition (LBC) fields diffusing (smoothing) the differences between the model solution and LBC. The sponge zone is characterized by a varying level of numerical instabilities. Coming to the simulations analyzed in the present study this buffer zone seems to be almost absent between the second (d02) and third (d03) nested domains. Subsequently, I expect a very noisy field driving d03 and I wonder if and to what extent this could negatively impact the proper development of d03 dynamics. Could the Authors provide any justifications for this quite atypical nesting strategy? And if any numerical detrimental effect has been detected or can be excluded.

A: The basic setup of the grid with a 1st (50 km) and 2nd (7 km) nesting step of the ensemble started more than 15 years ago when a resolution of 50 km was still common. It was originally designed to generate a 7 km ensemble (e.g. Feldmann et al. 2013). For consistency, nest 1 and nest 2 were kept constant for subsequent simulations where nest 2 was used to force the 2.8 km ensemble. However, the model version (to CCLM5) and forcing data (CMIP5 forcing) have been updated compared to the first simulations and are consistent throughout the nesting steps.

We agree that with higher resolutions a larger boundary zone between Nest2 and 3 would be advantageous to fully develop the fine scale features. In the present setup we could benefit from a step in horizontal resolution of less than a factor of 3 between Nest2 and Nest3, which is smaller than in common convection permitting setups today (Ban 2021). This is likely to reduce boundary effects in favour of a tighter nesting. However, the boundary zone that was truncated was quite large (48 grid

points, 137 km). An examination of the boundary effects in our results showed that anomalies in temperature and heat waves, as well as mean and extreme precipitation occur well outside the analysis domain. We will mention the nesting aspects in section 2.1 regarding the model setup.

- Feldmann, H.; Schädler, G.; Panitz, H.-J.; Kottmeier, C., 2013: Near future changes of extreme precipitation over complex terrain in Central Europe derived from high resolution RCM ensemble simulations. International journal of climatology, 33 (8), 1964–1977. doi:10.1002/joc.3564

- Ban, N., Caillaud, C., Coppola, E., Pichelli, E., Sobolowski, S., Adinolfi, M., Ahrens, B., Alias, A., Anders, I., Bastin, S. and Belušić, D., 2021. The first multi-model ensemble of regional climate simulations at kilometer-scale resolution, part I: evaluation of precipitation. Climate Dynamics, 57, pp.275-302.

**General Comment 2:**
My second concern regards the bias correction.

(i) from a technical point of view the quantile mapping (QM) configuration is not sufficiently described. I am especially referring to the correction of future time segments. As many studies point out empirical or parametric QM can affect the original climate change signal e.g., (Maraun 2016). So, it is a relevant choice to let QM free to alter the original simulated change signal, or conversely apply a trend-preserving QM configuration. Another relevant point is the extrapolation of the correction function over extreme values not present in the reference period but appearing in the future period.
(ii) I do not get the meaning of bias adjusting convection-permitting (CP) scale simulations. This is not in general as we know they are still to some extent affected by processes misrepresentation but in the context of this study. Firstly, it is not clear when and how bias-adjusted simulations are considered in the analyses. I suggest making much clearer this point throughout the manuscript. Further, how we can disentangle the so-called added value of CP-scale simulations generated by an improved representation of physical processes and what is generated by the application of bias adjustment if raw and adjusted simulations analyses are not compared? This is especially when the same time segment is considered for deriving QM correction function and to evaluate it since the adjusted simulations and the observations will have by construction very similar statistical moments. Since CP-scale climate simulations are only very recently affordable and many aspects are still to be explored (like mechanisms and dynamics underlying hot extremes) I would rather focus on exploring the eventual added value and weaknesses of original CP-scale simulations compared to the (original) non-CP-scale simulations. This is just a suggestion I am not asking to rewrite a new manuscript. Also, for what concerns future changes, I would rather be interested in the influence of the CP-scale on eventual trend modification instead of using bias-adjusted CP-scale simulations since this latter could have modified original trends as well, especially considering extremes. Here, it is complicated to isolate the "real" effect of the high-resolution shuffling bias correction in this context. I believe that the manuscript already rises many relevant points even without including bias correction since represents another layer of uncertainty over statistics and climate indices that represent a rather high level of sophistication,

even though not all the analyses involve bias-adjusted simulations (which increases confusion in the storytelling).

(i) A: We have revised the section and added the missing information:

*In order to correct for a systematic error in climate simulations to obtain reliable data for impact assessment, it is common practice to apply a bias correction (Maraun 2016). Following the assumption, that the model bias remains constant over time for each quantile of the model data, we apply quantile delta mapping according to Cannon (2015). Its application to a modeled variable $x_{mod,pred}$ at time step t in the prediction period (*pred*) is based on its non-exceedance probability $P_t$, which is evaluated over the cumulative distribution function F. A quantile mapping of the value with the same non-exceedance probability $P_t$ in the historical period (*hist*) is performed based on observed reference data (*obs*). To preserve the relative changes between the historical and the prediction period, the climate change signal $\Delta_m$ of the corresponding quantile is multiplied to obtain the corrected value $y_{mod,pred}$ (Eq. 2 & 3).*

$$P_t = F_{mod,pred}\left(x_{mod,pred}(t)\right) \qquad \text{(Eq. 1)}$$

$$\Delta_m(t) = \frac{x_{mod,pred}(t)}{F^{-1}_{mod,hist}(P_t)} \qquad \text{(Eq. 2)}$$

$$y_{mod,pred}(t) = F^{-1}_{obs,hist}(P_t) * \Delta_m(t) \qquad \text{(Eq. 3)}$$

*A normal distribution was fitted to the distribution of absolute temperature to derive the transfer function similar to Quian and Chang (2021). For the correction of precipitation the empirical approach is used in contrast, as no added value was found with the distribution-based method using e.g. a gamma distribution. In addition, a dry-day correction following Ehmele et al. (2022) was applied prior to the correction for precipitation.*

*The bias correction was derived for the parameters daily mean temperature $T_{mean}$, daily minimum temperature $T_{min}$, daily maximum temperature $T_{max}$, and the daily precipitation sum $P_{sum}$. As reference the observation dataset HYRAS with a horizontal resolution of 5 km was used, that was interpolated to the model grid. Along with the interpolation a height correction of $T_{mean}$, $T_{min}$ and $T_{max}$ was applied assuming a vertical gradient of 0.0065 K/m. The available 30 years of the historical time slice from 1971 to 2000 were used as reference period. To account for seasonal dependencies as discussed in Prierce et al. (2015), evaluation was done over a three month window. To minimize discontinuities at the edges of the time window (Prierce et al., 2015), the bias correction was applied for each month i of the year separately, using a transfer function derived and applied over month i-1 to month i+1.*

*This approach was chosen because it preserves the climate change signal of the quantiles, which is important for the relative description of heat waves used in the study. Furthermore, the method allows an application of the correction in a future climate where the temperature may exceed the range of temperatures in the historical period which is only possible to a limited extent with classical quantile mapping (QM) (Maraun 2016). However, the underlying assumption and the resulting constant transfer function might not be valid in a future climate (Prierce et al. 2015), leading to potential errors.*

- Maraun, D., 2016. Bias correcting climate change simulations-a critical review. Current Climate Change Reports, 2(4), pp.211-220.

- Cannon, A.J., Sobie, S.R. and Murdock, T.Q., 2015. Bias correction of GCM precipitation by quantile mapping: how well do methods preserve changes in quantiles and extremes?. Journal of Climate, 28(17), pp.6938-6959.

- Qian, W. and Chang, H.H., 2021. Projecting health impacts of future temperature: a comparison of quantile-mapping bias-correction methods. International Journal of Environmental Research and Public Health, 18(4), p.1992.

- Ehmele, F., Kautz, L.A., Feldmann, H., He, Y., Kadlec, M., Kelemen, F.D., Lentink, H.S., Ludwig, P., Manful, D. and Pinto, J.G., 2022. Adaptation and application of the large LAERTES-EU regional climate model ensemble for modeling hydrological extremes: a pilot study for the Rhine basin. Natural Hazards and Earth System Sciences, 22(2), pp.677-692.

- Pierce, D.W., Cayan, D.R., Maurer, E.P., Abatzoglou, J.T. and Hegewisch, K.C., 2015. Improved bias correction techniques for hydrological simulations of climate change. Journal of Hydrometeorology, 16(6), pp.2421-2442.

(ii) A: As the reviewer states, an analysis of the added value is only reasonable for uncorrected data. We assume that there is a misunderstanding here, as the analysis in section 3 is carried out with the raw data of the model without correction. A correction is only applied in the following sections 4 (analysis of regional temperature trends), 5 (heat wave characterization) and 6 (impacts of temperature and heat increase). We agree with the reviewer that caution is needed with the bias correction. We chose to bias-correct despite this, as our focus is on relating our results to impacts and thus the application of climate data, and try to go beyond an evaluation of the convection permitting dataset.

We will revise the manuscript to clarify when the bias-corrected simulations are used and when the uncorrected model data are used. We hope this clarification resolves the reviewer's concern.

Specific comments

Line 4. "We find an improved mean temperature beyond the effect of a better representation of orography on the convection-permitting scale, with reduced bias, particularly during summer". I do not believe that the manuscript analyses allow us to reach such a conclusion.

A: Following the general comment 2, we will clarify that the evaluation of the added value is based on uncorrected simulation data. We hope that this clarification resolves the reviewer's concern.

Moreover, we have clarified the statement to:

*"We find a systematically reduced cold bias especially in summer in these simulations compared to the driving simulations with a grid size of 7 km and parametrized convection."*

The caption of Figure 1. To me it results quite complicated to understand, please rephrase. Especially: "Nesting in (a) and model domain"

A: Changed to: *"In (a) the three nesting levels are shown. (b) shows the model domain with the sponge area truncated and the used evaluation area in red."*

Section 2.2 should be improved (see general comments.).

A: See above

Line 152. "user-oriented parameterizations are tested". Please explain what you mean in this statement.

A: We have clarified the sentence to: *"Finally, climate parameters - threshold based indices that are tailored to the need of practice stakeholders in different fields action - are evaluated."*

A detailed description follows in the method section.

Lines 172-174. To me is not cleat the meaning of "reference humidity is constant at 20hPa" is. Please clarify.

A: We have clarified the paragraph to: *"The relative humidity in the reference environment is 50% for temperatures below 29°C. However, for temperatures above 29°C, the water vapour pressure is instead kept constant at a level of 20hPa."*

Line 187. Please correct the quote's typo.

A: The typo has been corrected.

Figure 3. Instead of monthly means, I would rather compare the five daily temperature distributions (e.g., boxplots) or a percentile-based error to see which part of the distribution benefits the most from the higher resolution during the different parts of the year.

A: We appreciate the suggestion and have evaluated the distributions of temperature for the months. We have updated Figure 3a (see Figure) and added the 10th and 90th percentile to the graph. Further percentiles were evaluated with consistent results and therefore not added to the graph.

*We would add following paragraph in the manuscript:*

*Figure 3a shows that in the reanalysis driven simulation the median monthly temperature over the evaluation domain in the 7 km simulation (blue thick solid line) is always lower than in the observation (gray thick solid line). This deviation is larger in the summer months. A similar pattern is found for further percentiles of the distribution, as shown for example for the 10th and 90th percentiles (thin lines in Fig. 3a), as they are generally underestimated, especially in summer. However single autumn and winter months occur (October for the 90th and January for the 10th percentile) where the 7 km output exceeds the observation. In the convection permitting simulation (2.8 km), the monthly median temperature in the warm season is comparably higher than in the coarser simulation, leading to a reduced cold bias. In autumn it even exceeds the observation by 0.6 K. However, there is no strong improvement in the mean temperature during the winter months and the cold bias persists. A consistent reduction of the cold bias is found for the 10th and 90th percentiles, but possible overestimation of higher percentiles seems to become more frequent especially in late summer and autumn.*

[Figure]

Figure 3a: (a) shows the monthly mean temperature in the observation (black solid lines) compared to the reanalysis results (colored solid lines) and the median of the ensemble members (dashed lines). The thick line represents the median in the reference period and in the evaluation area, the thin lines show the 10th and 90th percentiles respectively.

Line 208. Please clarify how the Wilcoxon test is applied in this context.

A: The Wilcoxon signed-rank test was applied to the two fields of mean error of the coarser 7km simulation and the convection-permitting 2.8km simulation. The null hypothesis was that the difference in the mean error of coarse and fine grid is zero. The null hypothesis was rejected by the test.

We have clarified this in the manuscript.

Line 210. Why is talk about trends here? This sentence is not clear to me.

A: We have changed the wording to "*patterns*".

Figure 4 caption. (c) appears twice.

A: We have corrected the caption.

Line 239. "Average variance" Perhaps ensemble variance?

A: To clarify the sentence without loosing the information about spatial averaging, we have changed the sentence to *"There is an ensemble variance of 0.6 $K^2$ for the mean temperature averaged over the study area"*

Line 252. Please clarify the meaning of "the full width of half maximum (FWHM)".

A: We have added the definition and meaning of the full width at half maximum in relation to the changes in temperature distribution shown:

*A parametrization of the spread of the distribution is made in terms of the Full Width at Half Maximum (FWHM), which is defined as the width of the distribution at the level of the half peak value. [...] Regarding the temperature distribution, an increasing FWHM indicates a more variable daily*

*temperature, leading to higher amplitudes and to a stronger increase in the frequency of warm extremes on the right side of the curve compared to the shift of the curve median.*

Figure 6. Why change color bar limits and colormap between (a) and (c) panels?

A: A common color range did not allow to visually distinguish regional differences. Therefore, two different colobars were introduced, each "zooming" to the area of the change signal in GWL2 and GWL3. The color scheme was also changed to emphasize the different range as well as different scaling.

Heat wave characterization results section is quite loosely described, I would suggest better discussing this part.

A: We will revisit the section and strengthen the discussion in a revised version.

Figure 7. caption, the description of the panel (d) is not clear to me. Also, PDFs are not described.

A: We would clarify clarified the description to: *"The bubble plots show the strongest HW in each summer half-year (May-Oct) in every projection run with respect to duration on the abscissa and excess temperature on the ordinate. Bubble size indicates mean HWMId over all grid point results affected by the HW. Marginal plots show the distribution of duration of the heat waves in days (abscissa) and the distribution of the excess temperature (ordinate). [...] Panel d shows the total set of all of the heat waves from the single ensemble members for 1971-2000, GWL2 and GWL3. [...]"*

Discussion and conclusion section. Here it should be clarified how the CP-scale and/or the bias correction contribute to the reported improvement of temperature extremes representation.

A: We hope that most of the reviewer's concerns are addressed by the above clarification that Section 3 is based on uncorrected data. We will discuss the improvements of the chosen data and methods in more detail in the discussion and conclusion section. It was shown that the 2.8km simulations have a lower bias especially in summer (Fig. 3). The bias correction improves especially the representation of threshold based parameters (e.g. Fig. 11). It reduces the effect of model biases on the ensemble spread, which then more defined by the (regional) climate sensitivity of the selected GCMs.

---

## Author Comment (AC2)

**Anonymous Referee 2:**

The present study is dedicated to summer temperature, heat waves and associated implications for human health, agriculture and tourism in an ensemble of convection permitting regional climate model projections. In addition, the added value of the higher model resolution is demonstrated compared with a model version using parameterized convection. The study comprises three novel aspects at once: (1) a relatively large model data base of very high-resolution simulations over a quasi-transient forcing period, (2) the assessment of regional to local climate change patterns based on substantially improved model simulations, and (3) the consideration of derived climate indicators bridging the gap between meteorological heat events and socio-economic implications as well as adaptive requirements.

The paper represents a very valuable contribution to the community – with respect to methodical aspects (new model generation) and practice-relevant research (high-resolution patterns of climate change). Therefore, I recommend this manuscript to be accepted for publication in NHESS with minor revisions.

The minor revisions refer to a list of specific comments (see below) and to two general comments:

A: First, we would like to thank the reviewer for her/his insightful comments, which have greatly contributed to improving the text. In making corrections, we have tried to follow the suggestions as closely as possible.

**General comments**

**General comment 1**

(1) The manuscript basically is well presented, but exhibits some linguistic inaccuracies, especially typos. Therefore, I believe the authors themselves can achieve an improvement without explicit language editing by a native speaker. Nonetheless, a careful revision is required since the typos and inaccuracies are quite numerous.

A: We appreciate the feedback and will revise the manuscript.

**General comment 2**

(2) The GWL 2 and 3 periods seem to be associated with a lower level of temperature increase in central and southern Germany, at least in terms of the mean summer temperature. According to the IPCC and many other studies, I would have expected an above-average warming in Central Europe, given the fact that land masses are warming up stronger than the ocean surface, especially in the Northern Hemisphere extratropical regions (COWL pattern). Is summer less sensitive than the annual mean or is it an issue of the considered GCM-RCM combinations? I suggest that the authors pick up this point in their discussion.

A: Indeed, the warming is slightly stronger integrated over the year. We have rewritten the respective paragraph:

*The summer temperature increases with global warming over the whole evaluation area. From the reference period (global warming at 0.46 °C) to GWL2, the increase is on average 1.55 °C (Fig. 6a). From the reference period to GWL3 the average increase is 2.60 °C. When integrated over the year, the*

*ensemble shows a slightly stronger warming than only over the summer months, indicating that summer temperatures are less sensitive than the annual mean (Fig. 5a). However, the differences are still in the range of 0.11 °C (0.09 °C) above the global warming in GWL2 (GWL3). Therefore, the regional warming in the evaluation area in the considered GCM-RCM combinations is close to the global average and only slightly enhanced. This is less than suggested by the theory of greater warming over land than over the ocean and as generally projected (IPCC 2021). The impact of the bias correction on the climate change signal is considered to be negligible, as the uncorrected data integrated over the year show a nearly identical warming of 0.11 °C (0.07 °C) above the global average in GWL2 (GWL3) in the evaluation area.*

*Geographical dependence leads to regional variations of warming. Over the evaluation area, warming ranges from 1.45 to 1.64 °C (5th And 95th percentiles) in GWL2 and from 2.44 to 2.76 °C in GWL3. As shown in Fig. 6a and 6c, the strongest increase is observed in the uplands in the north of the domain (GWL2 ), and in the Black Forest and Swabian Alps in the south (GWL2 and GWL3). Less warming, below the global average, is expected in the Alpine Foreland  (GWL2 and GWL3).*

To fit in the new structure of the section a following paragraph about the ensemble spread was rearranged as well.

- IPCC: Climate Change 2021: The Physical Science Basis. Contribution of Working Group I to the Sixth Assessment Report of the Intergovernmental Panel on Climate Change, Cambridge University Press, Cambridge, United Kingdom and New York, NY, USA, https://doi.org/10.1017/9781009157896, 2021.

**Specific Comments:**

Line 42: CPM stands for 'convection permitting model' (not convective).

A: We have corrected the typo.

Line 62: What is meant by quasi-transient? And 'manor' is certainly not the right word in this context, I guess it is 'manner'.

A: We have corrected the typo. We have added the following in the introduction, a detailed description is available in the methods section:

*All simulations cover the period from 1971 to 2100 in a quasi-transient manner, where the projection is composed of several time slices.*

Line 85: Table 2 is addressed in the text before this is done for Table 1.

A: We have corrected the Table's location and reference.

Fig. 1: The fine lines in the background of the map seem to be river basin. Maybe a word is useful why these are plotted.

A: The lines in the background represent the German major landscapes. Those regions were added, because often a dependency of the results is visible and they facilitate the interpretation of the results. We have added a description of this background map in the figure description.

Line 96: I wouldn't call it a climatological difference, when two three-year periods are compared with each other. Maybe the authors may want to call it what it is actually: a difference between three-year averages.

A: We have clarified the sentence as suggested.

Subsection 2.1: I suggest to explain in few words the data sources and procedure leading to the HYRAS dataset and to explain what an equilibrium climate sensitivity is (Table 2).

A: We have added a short explanation of equilibrium climate sensitivity.

*Regarding HYRAS, we have added the following: "The HYRAS dataset is used as observation, which is based on station data that are aggregated using the REGNIE method of combining a regression model and inverse distance weighting to a gridded dataset (Rauthe et al. 2013, Razimaharo et al. 2020)."*

- Rauthe, M., Steiner, H., Riediger, U., Mazurkiewicz, A., and Gratzki, A.: A Central European precipitation climatology–Part I: Generation and validation of a high-resolution gridded daily data set (HYRAS), Meteorol. Z, 22, 235–256, https://doi.org/10.1127/0941-2948/2013/0436, 2013.
- Razafimaharo, C., Krähenmann, S., Höpp, S., Rauthe, M., and Deutschländer, T.: New high-resolution gridded dataset of daily mean, minimum, and maximum temperature and relative humidity for Central Europe (HYRAS), Theor. Appl. Climatol., 142, 1531–1553, https://doi.org/10.1007/s00704-020-03388-w, 2020.

Line 131: As this study is focussed on heat events, the question arises whether extreme temperature is indeed normal. There are several studies indicating that it is not, suggesting a combined QM approach with different statistical models below and above a temperature threshold. Please add a discussion on this issue.

A: We would rewrite the section on bias correction, also in light of the comments in Review 1 (see response to Review 1). We would add a discussion off the distribution based approach in a revised manuscript.

*Furthermore, the use of a parametric approach of fitting an assumed distribution to the data to derive the transfer function is still arbitrarily discussed. Several studies, e.g. Pastén-Zapata et al. (2020), Quian et al. (2021), apply a normal distribution for temperature to get a more robust transfer function. Using a fitted function has the additional advantage that the transfer function is independent of any smoothing interval that may be defined (Kerkhoff et al. 2014). On the other hand parametric approaches introduce additional bias, if the distribution of a variable is not accurately met by the theoretical distribution. Especially for extreme values, a deviating statistic is assumed according to the extreme value distribution. Quantile approaches, allowing different statistical models for extremes,*

*could potentially reduce uncertainty (e.g. Vrac and Naveau 2007, Berg et al. 1012, Schubert et al. 2017).*

- Pastén-Zapata, E., Jones, J.M., Moggridge, H. and Widmann, M., 2020. Evaluation of the performance of Euro-CORDEX Regional Climate Models for assessing hydrological climate change impacts in Great Britain: A comparison of different spatial resolutions and quantile mapping bias correction methods. *Journal of Hydrology*, *584*, p.124653.

- Qian, W. and Chang, H.H., 2021. Projecting health impacts of future temperature: a comparison of quantile-mapping bias-correction methods. *International Journal of Environmental Research and Public Health*, *18*(4), p.1992.

- Kerkhoff, C., Künsch, H.R. and Schär, C., 2014. Assessment of bias assumptions for climate models. *Journal of Climate*, *27*(17), pp.6799-6818.

- Berg, P., Feldmann, H. and Panitz, H.J., 2012. Bias correction of high resolution regional climate model data. *Journal of Hydrology*, *448*, pp.80-92.

- Schubert, D., van der Linden, R., Fink, A.H., Katzfey, J., Phan-Van, T., Maßmeyer, K. and Pinto, J.G., 2017. Klimaprojektionen für die hydrologische Modellierung in Südvietnam. *Hydrologie und Wasserbewirtschaftung*, *61*(6), pp.383-396.

Subsection 2.3.2: The description of UTCI is deficient. I either suggest to refer to the literature, leaving out all equations, or to provide a complete description with all terms figuring in the equations and the full equation for UTCI instead of f().

A: The complexity of the overall calculation is beyond the scope of this paper, so we have decided to refer to the literature.

Line 207: It should be mentioned that this statement refers to the reanalysis-driven experiment. The enhanced spread is probably related to the fact that the model has a higher genuine resolution than the validation data, implying higher temperature differences in mountainous areas.

A: We have added that information.

Fig. 4: Panel c is unclear to me: is it a comparison of the bias (then the caption is wrong saying that it is the 2.8 km minus 7 km scale) or does it indicate that the negative bias of the 7 km run is more or less compensated by the 2.8 km run. I would prefer seeing a bias reduction in panel c because it is more intuitive for the reader.

A: The mean square error skill score (MSESS) is displayed (Eq. 3) in c. We have corrected an error in describing the labeling in the caption. We hope that this resolves the reviewers concern.

Beginning of section 4: I miss a statement about what model resolution is used for the subsequent analyses. I guess it is the 2.8 km scale since the bias could be reduced noticeably.

A: This is correct, the 2.8km resolution was used in the following. We have added a statement regarding the used resolution in Section 4 to 6.

Line 247: Have the authors tested whether the density is indeed skewed left. At first sight, it looks quite normal.

A: That is a valid point. We have recalculated the skewness. In the reference period (1971-2000) it is between -0.24 and -0.18 in the ensemble, while in the observation it is -0.17. Since these deviations from 0 are relatively small, we have removed this aspect from the manuscript.

Line 252: Please explain what FWHM actually tells us.

A: We have added the definition and meaning of the full width at half maximum in relation to the changes in temperature distribution shown:

*A parametrization of the spread of the distribution is made in terms of the Full Width of Half Maximum (FWHM), which is defined as the width of the distribution at the level of the half peak value. [...] Regarding the temperature distribution, an increasing FWHM indicates a more variable daily temperature, leading to higher amplitudes and to a stronger increase in the frequency of warm extremes on the right side of the curve compared to the shift of the curve median.*

Line 277: At the end of this sentence the authors may include a '(not shown)'.

A: Line 277 comprises following statements "*Overall, the mean temperature over Germany rises in a warmer climate predominantly in late summer as well as in the winter half-year, with the smallest increase in spring. This leads to a general shift of the summer maximum temperatures to later summer*" The paragraph serves as a summary of the analysis above. The statement in line 277 was discussed in line 234 to 237 based on the evaluation presented in Fig. 5a. We have clarified that the paragraph is intended as a summary and hope that this addresses the reviewer's concern.

Fig. 9: It should be mentioned that the thick solid line refers to the ensemble mean. To be clear please add 50% 'of the study region'.

A: We have changed the description as suggested.

Line 357: Why is the British model now claimed an outlier whereas previously it was not because blocking situations may be better represented in this model?

A: In section 5, we found good agreement of the analysis of the bias corrected data of all 4 simulations, including HadGEM2-ES, with the present day conditions derived from the observation. In the following, we found particularly long heat waves in a future climate in the simulation driven by HadGEM2-ES. Those long, and therefore persistent, warm spells could originate from a different representation of large scale circulation patterns in the driving GCM, namely blocking situations.

In Section 6, we proceed analogously and first evaluate the present day conditions in all simulations. However, compared to the other 3 simulation chains, we found a significantly higher UTCI in the simulation driven by HadGEM2-ES. We therefore consider these results to be an outlier. We attribute this difference primarily to the fact that unlike the analysis in Section 5, no bias correction was applied to the hourly data used in Section 6.

We would add the following explanation to the respective paragraph:

*There is good agreement between three of the four ensemble members, showing a similar range of UTCI over the reference period 1971-2000. The simulation driven by HadGEM2-ES results in a significantly higher number of days with UTCI > 32°C. We attribute this difference mainly to higher summer temperatures in this simulation, which unlike the previous analysis of daily data was not subject to bias correction. To minimize the influence of possible outliers, we consider the ensemble median in the following analysis.*